# Effect of Endogenic and Exogenic Oxidative Stress Triggers on Adverse Pregnancy Outcomes: Preeclampsia, Fetal Growth Restriction, Gestational Diabetes Mellitus and Preterm Birth

**DOI:** 10.3390/ijms221810122

**Published:** 2021-09-19

**Authors:** Eun Hui Joo, Young Ran Kim, Nari Kim, Jae Eun Jung, Seon Ha Han, Hee Young Cho

**Affiliations:** 1Department of Obstetrics and Gynecology, CHA Bundang Medical Center, CHA University, Seongnam-si 13496, Korea; a186023@chamc.co.kr (E.H.J.); happyiran@chamc.co.kr (Y.R.K.); a186053@chamc.co.kr (N.K.); a196058@chamc.co.kr (J.E.J.); a216011@chamc.co.kr (S.H.H.); 2Department of Obstetrics and Gynecology, CHA Gangnam Medical Center, CHA University, Seoul 06135, Korea

**Keywords:** oxidative stress, preeclampsia, fetal growth restriction, preterm birth, gestational diabetes mellitus

## Abstract

Oxidative stress is caused by an imbalance between the production of reactive oxygen species (ROS) in cells and tissues and the ability of a biological system to detoxify them. During a normal pregnancy, oxidative stress increases the normal systemic inflammatory response and is usually well-controlled by the balanced body mechanism of the detoxification of anti-oxidative products. However, pregnancy is also a condition in which this adaptation and balance can be easily disrupted. Excessive ROS is detrimental and associated with many pregnancy complications, such as preeclampsia (PE), fetal growth restriction (FGR), gestational diabetes mellitus (GDM), and preterm birth (PTB), by damaging placentation. The placenta is a tissue rich in mitochondria that produces the majority of ROS, so it is important to maintain normal placental function and properly develop its vascular network to ensure a safe and healthy pregnancy. Antioxidants may ameliorate these diseases, and related research is progressing. This review aimed to determine the association between oxidative stress and adverse pregnancy outcomes, especially PE, FGR, GDM, and PTB, and explore how to overcome this oxidative stress in these unfavorable conditions.

## 1. Introduction

Pregnancy is a natural process in women, that involves a series of complex events, including implantation, placentation, and childbirth. In this series of events, placentation is quite important because of the various functions it performs and because any disruption within it could lead to serious complications for the fetus and the mother. The placenta provides fetal oxygenation and nutrition. During oxygenation and deoxygenation of the placenta, some by-products of oxygen, that is reactive oxygen species (ROS), are inevitably formed. Oxidative stress can be generated upon disruption of the balance between ROS formation and detoxification. These by-products are known to induce inflammatory responses and damage the cellular system, even at the DNA and RNA levels, leading to premature placental aging. Premature aging of and degenerative changes to the placenta may decrease its functional capacity and lead to abnormal pregnancy outcomes such as preeclampsia (PE), fetal growth restriction (FGR), gestational diabetes mellitus (GDM), and preterm birth (PTB) [1,2]. In addition, if DNA is damaged by oxidative stress or any other problems, then the ovum or sperm become unavoidably mutated or undergo apoptosis, resulting in infertility, impaired embryogenesis, or pregnancy-related diseases [3].

Therefore, it is important to maintain the appropriate ROS levels, but our modern lifestyles lacking adequate physical activity and exposure to various toxic substances disturb these significant processes. Our bodies gradually became more susceptible to increased inflammation and oxidative stress. This eventually leads to a higher risk of molecular system damage. Reactive nitrogen species (RNS) can also influence pregnancy. Thus, fetal development and subsequent adult diseases are related to oxidative stress [4].

Several studies have discussed the influence of oxidative stress in pregnancy and related complications in fetuses and mothers [5,6,7]. Interestingly, in the early stages of placental development, a relatively low oxygen concentration protects early embryos from oxidative stress. However, once the placental development is complete, its oxygen level has risen almost three-fold compared to that of the initial stages to ensure an adequate oxygen supply to the fetus [8]. It is important to maintain a well-controlled oxygen level, during each period of pregnancy. Antioxidant activity is also important for maintaining stable oxygen levels in the placenta. A depletion of antioxidant capacity through a low abundance of enzymatic or non-enzymatic antioxidants makes the cells vulnerable to oxidative stress, even under physiological conditions in which the redox status is maintained through a careful balance of a low ROS synthesis level and the pathways of cellular defense [9]. Antioxidant therapy, which includes vitamins A, C, and E, selenium, folic acid, and flavonoids, may ameliorate or prevent disease [10,11,12].

In this study, we attempted to determine the relationship between ROS and its damaging effect on placentation, among other molecular influences that can harm the normal pregnancy progress. Here we discuss the placenta-mediated adverse outcomes of pregnancy associated with oxidative stress, particularly PE, FGR, GDM, and PTB.

## 2. Oxidative Stress and Pregnancy

Oxygen is the final acceptor of the electrons generated during cellular metabolic activities. These activities primarily include those of oxidases (xanthine oxidoreductase, NADPH oxidases), nitric oxide synthase (NOS), and mitochondrial oxidative phosphorylation. Di-oxygen goes through a tetravalent reduction and combines with two protons to produce water (H₂O) under normal physiological conditions. However, under pathological conditions, oxygen may be incompletely reduced to another form, which we call ROS. There is a well-equipped antioxidant system in the human body that protects us from the harmful effects of oxidative free radicals. With the disruption of redox homeostasis, the balance between antioxidants and ROS, oxidative stress can occur [13].

When a woman becomes pregnant, the placenta is created in her uterus to ensure sufficient fetal oxygen and nutrition. The trophoblast cells of the placenta gradually invade the myometrium and replace endothelial and smooth muscle cells in the uterine spiral arterioles. This process is crucial for increasing utero-placental blood flow during a normal pregnancy [8]. Implantation involves the interaction between several vasoactive agents, including cytokines, prostaglandins, and nitric oxide (NO∙), leading to an increase in matrix metalloproteinases. Indeed, NO plays a key role in decidualization and embryo implantation, which increases vascular permeability, vasodilation, and blood flow in the uterus. Some experimental studies reported that when NO was inhibited in the rat decidua, the apoptotic rate of decidual cells increased, suggesting that NO plays an important role in the survival of these cells [14].

There are many explanations of the hazardous roles of oxidative stress during placentation. One is that the impairment of antioxidant activity during placentation may lead to an increase in lipid peroxidation and subsequent vascular endothelial damage [15,16]. Another explanation is that low-grade ischemic reperfusion injury occurring secondary to abnormal remodeling of the spiral arteries during placentation may cause oxidative stress [17,18]. Some studies reported the association between hypoxic damage and apoptosis and that cultured trophoblasts exposed to hypoxia showed distinct upregulation of the p53 tumor suppressor protein enhanced expression of the pro-apoptotic Mtd-1, and decreased expression of the anti-apoptotic Bcl-2 protein, all of which promote apoptosis [19,20,21]. Additionally, the increased apoptosis due to upregulated p53 and decreased Bcl-2 in placental syncytiotrophoblasts is associated with pregnancy pathologies including FGR and PE [22,23].

Interestingly, contrary to these ideas, oxidative stress may play a physiological role in proper placentation. As mentioned above, early placental development occurs in a relatively low-oxygen environment to protect the early embryo from ROS; once utero-placental circulation has been established, a three-fold increase in oxygen concentration occurs within the placenta. This leads to elevations in ROS concentration in the syncytiotrophoblast, which has a low concentration of enzymatic antioxidants, and appears to preferentially trigger an apoptotic cascade in the peripheral villi and help the placenta regress into its definitive discoid shape [24,25]. Importantly, samples taken from the peripheral villi at this time had elevated levels of heat-shock protein 70, part of the heat-shock chaperone protein family, that are upregulated during oxidative stress. It is also involved in the folding and refolding of aggregated or misfolded proteins as well as nitro-tyrosine residues, indicative of peroxynitrite formation [26]. ROS and programmed apoptosis play an important role in maintaining homeostasis of the uterine endometrium during embryonic implantation, while elevated superoxide levels are believed to play a role in increasing vascular permeability during implantation. Finally, enzymatic and non-enzymatic antioxidants have been implicated in rescuing the corpus luteum, which is responsible for producing steroid hormones required for early placental maintenance from ROS-induced attacks [27,28].

Next, we will discuss the main sources of endogenous or exogenous ROS [1,29] and four major complications that are related to placental dysfunction, namely PE, FGR, GDM, and PTB.

### 2.1. Endogenous Sources

Our bodies have many sources of ROS generation in intracellular compartments. Intracellular compartments, including the mitochondria, endoplasmic reticulum, plasma membranes, and peroxisomes, participate in ROS production. Mitochondria, which have an electron transport chain system, are the main endogenous source of ROS in most mammalian cells. According to the free radical or oxidative stress hypothesis of aging, a number of events occur during oxidative phosphorylation resulting in the leaching of electrons from mitochondrial membranes that react with oxygen, forming free radicals such as superoxide and ultimately impairing the redox balance [30,31].

Specific enzymes, including peroxisomes, nicotinamide adenine dinucleotide phosphate (NADPH) oxidase, NADPH oxidase isoforms, xanthine oxidase (XO), lipoxygenase, glucose oxidase, and nitric oxide synthase, catalyze the ROS-generating chemical reaction [32].

### 2.2. Exogenous or Environmental Sources

Oxidative stress can also be generated by triggers outside the body. For example, air pollutants, tobacco smoke, alcohol, drugs, radiation and other environmental factors are also included. Chemical agents, heavy metals, organic solvents, and even pesticides, exposure to which often occurs in daily life, are also common exogenous sources of ROS. These factors irritate the body’s homeostasis, stimulate ROS formation and disturb the detoxification mechanisms of oxidative stress [32].

### 2.3. Measurement of Oxidative Stress

It has become popular among scientists and clinicians to identify the associations between oxidative stress level and perinatal outcomes to better understand the pathophysiology of complications that might be associated with pregnancy. However, a variety of methods are used to quantify oxidative stress, and it is difficult to draw a consistent interpretation and design new experiments.

Oxidative stress can be measured in three major ways: the direct measurement of ROS levels; the indirect measurement of protein, lipid, and DNA damage instead of assessing oxidative stress; the assessment of antioxidant status, which can be an indirect method for measuring oxidative stress [13].

Many of the oxidative stress measurements found in the literature are categorized as biomarkers, which, as described by the National Institutes of Health, are objective measurements or evaluations of biological processes (Biomarkers Definitions Working Group 2001) [33]. As previously mentioned, an imbalance between ROS formation and reduction damages lipids, proteins, and DNA. Thus, it is reasonable to evaluate these by-products from lipid, protein, and DNA damage as alternatives for the degree of oxidative stress.

Moore et al. [34] reported that enzyme-linked immunosorbent assay is most commonly used in PTB cases to identify the relationship between oxidative stress and total oxidant/antioxidant status. They also stated that the measurement of lipid peroxidation is also common. Although these methods are cost-effective and easily accessible, their results are very non-specific and extremely difficult to understand, even for experts. In addition, they are unable to identify the specific ROS and RNS or antioxidants involved in the processes for mechanistic interpretation.

Assessing antioxidant level/activity is another common way to measure oxidative stress [35,36]. Measuring antioxidant levels is more effective than measuring oxidative stress biomarkers because the results allow for further understanding of their potential mechanisms and their possible use as therapeutic interventions. However, it is essential to measure multiple antioxidants and include measures that identify specific ROS or RNS that may be associated with changes in antioxidant levels or activities to enable a more comprehensive understanding of the biological processes involving antioxidants.

Unfortunately, the reference values of ROS and NOS and even their minimal and safe concentrations or physiologically beneficial concentrations in pregnant women remain to be determined. Researchers should evaluate patients according to their etiological factors and perform individual analyses that include lifestyle factors such as diet and physical activity level. Additionally, an investigation of patients’ medical history according to biological samples collected before delivery, abortion or stillbirth and fetal health should be implemented. The measurement of oxidative stress in vivo is a debatable issue, as the sensitivity and specificity of various oxidative stress markers remain uncertain and a well-controlled laboratory environment is required to ensure their precision. Moreover, there might be interobserver differences in estimated values.

## 3. Oxidative Stress and PE

PE, a hypertensive disorder that is among the most severe and life-threatening complications of pregnancy, can be characterized by elevated blood pressure (≥140/90 mmHg), proteinuria, and evidence of end-organ damage, such as liver, kidney, and central nervous system dysfunction, a low platelet count, and pulmonary edema. If these problems are not corrected at an early stage, serious consequences can occur, such as brain damage (seizures, stroke, etc.) and other multiple adverse complications in various organs such as kidney and liver [37]. Although the etiology of PE remains debatable, its basic pathology is thought to be vascular endothelial injury mediated by oxidative stress from increased placental ROS or decreased antioxidant activity.

Polyunsaturated fatty acids, which are found in abundance in the cell membrane and circulating lipoproteins, are highly susceptible to oxidation by free radicals to lipid peroxide, a process called lipid peroxidation [38]. A normal pregnancy involves increased free radical production and lipid peroxidation; however, antioxidant activity is also upregulated to counterbalance them [39]. In contrast, PE is associated with increased lipid peroxidation in the maternal circulation and placenta and decreased antioxidant activity [38,40,41]. Consequently, trophoblastic invasion of the spiral arteries is inhibited, which limits spiral artery remodeling to the decidual portions, while the myometrial segments of the arteries remain narrow and contractile. Therefore, in PE, increased vascular resistance in the placenta results in reduced uteroplacental perfusion [2]. PE is also associated with FGR, PTB, and placental abruption due to abnormal placentation. [42,43].

In PE, the circulating and placental tissue levels of oxidative stress markers are elevated and antioxidant capacities are compromised [9,44]. Two major end products of lipid peroxidation, malondialdehyde (MDA) and 4-hydroxynonenal (HNE), are frequently used as indicators of lipid peroxidation and oxidative stress. Placental and serum levels of MDA and, 4-HNE and placental XO expression are increased in preeclamptic versus normotensive pregnancies, whereas maternal circulating and placental levels of antioxidants, such as catalase (CAT), glutathione peroxidase (GPX), and superoxide dismutase (SOD) are decreased in preeclampsic versus normotensive pregnancies [40,45]. During labor, the placenta suffers from periodic ischemia and reperfusion, leading to elevated oxidative stress markers and alterations in gene expression. Oxidative stress in the placenta induces the release of cytokines, angiogenic factors, and apoptotic debris into the maternal circulation, which can induce a series of inflammatory responses. Many studies have reported relatively higher oxidative stress markers levels in pregnancies with PE, a representative disease related to abnormal placentation [46]. In addition, when pregnant women develop PE, the levels of specific related markers, such as soluble fms-like tyrosine kinase (sFlt), vascular endothelial growth factor (VEGF), and placental growth factor (PlGF), also change. These markers are usually associated with angiogenic growth of the placenta and help clinicians predict disease earlier and prevent severe complications [47,48].

## 4. Oxidative Stress and FGR

FGR, also known as intrauterine growth restriction, is failure of the fetus to reach its genetic growth potential. Defined as an estimated fetal weight less than the 10th percentile for gestational age, FGR is a leading cause of fetal, neonatal, and perinatal mortality and morbidity [49]. Some potential risk factors for FGR include maternal smoking, infection, obesity, malnutrition, and chromosomal abnormalities, but the majority of cases remain unexplained [50]. The most common etiology of FGR is uteroplacental dysfunction, which leads to diminished maternal uteroplacental blood flow [51]. It was recently hypothesized that placental insufficiency originates in early gestation when the trophoblast invades the spiral arteries in the placental bed [52]. This process requires high energy availability for cell growth, proliferation, and metabolic activity, which generates ROS and oxidative stress. Adequate trophoblastic invasion of the spiral arteries may occur when the chorioallanotic villi encounter an injury caused by stimuli or mediators [53]. Among the various stimuli or mediators, oxidative stress plays a leading role. Consequently, incompletely developed spiral arteries cause ischemia-reperfusion, which exacerbates oxidative stress and contributes to damaging the placental tissue.

Damage resulting from oxidative stress predominantly occurs in membrane lipids, proteins and nuclear and mitochondrial DNA. Plasma and tissue levels of MDA, an end product of fatty acid oxidation, are frequently measured as lipid peroxidation and oxidative stress indicators. MDA and XO levels are higher in the maternal plasma, umbilical cord plasma, and placental tissues of patients with FGR pregnancies than in those with healthy pregnancies [54], suggesting that oxidative stress plays a role in FGR.

Moreover, the FGR placenta shows signs of aging markers, including telomere shortening and absent or reduced telomerase activity [55,56]. In addition, the expressions of telomere-induced senescence markers p21 and p16 are elevated, while levels of the anti-apoptotic protein Bcl-2 are decreased in the FGR placenta [55,57,58]. Together with increased oxidative stress markers and reduced antioxidant capacity, this evidence of aging markers supports the concept that oxidative stress plays a role in placental aging and FGR.

## 5. Oxidative Stress and GDM

In GDM, a hormone from the placenta prevents the body from using insulin effectively. GDM is a heterogeneous disorder involving a combination of factors responsible for decreased insulin sensitivity and inadequate insulin secretion. The underlying pathophysiology of GDM is, in most instances, similar to that in type 2 diabetes. The inability of pancreatic beta cells to match the increased insulin resistance to normalize the systemic glucose level translates to maternal hyperglycemia. Similar to type 2 diabetes, GDM is a multifactorial disease associated with both genetic and environmental risk factors [59].

It is widely known that a hyperglycemic environment is associated with oxidative stress. Therefore, pregnancy accompanied by GDM is associated with increased oxidative stress levels compared to normal pregnancy. Indeed, ROS overproduction occurs and free radical elimination mechanisms are impaired in women with GDM. This defective antioxidant system can lead to embryonic and fetal exposure to the harmful effects of oxidative stress. It is also associated with suboptimal decidualization of the placenta [60]. There is a higher incidence of congenital malformations in the offspring of diabetic women, and some evidence suggests that higher lipid peroxidation levels and lower antioxidants levels may be causative factors [61]. Women with GDM are also at an increased risk for complications such as endothelial dysfunction and cardiovascular diseases [62]. It has been shown that lipid profile controlling during GDM can prevent impairment of the feto-placental endothelial function [63].

There is certainly much evidence to suggest that, in GDM pregnancies, placental production and antioxidant enzyme activities increase to maintain redox homeostasis. However, catalase activity is reportedly decreased in the placenta of women with GDM [64]. In human studies, in the placenta from experimental models of GDM, antioxidant enzymes were either upregulated to compensate for oxidative dysbalance or downregulated by the increased ROS levels. These changes are dependent on developmental stage and generated in response to the gradual increase in ROS level, which is more pronounced at pregnancy term [65,66].

## 6. Oxidative Stress and PTB

PTB, defined as delivery prior to 37 weeks of gestation, is the leading cause of morbidity and mortality in neonates, affecting approximately 10% of newborns in the United States. PTB can be divided into three groups: medically induced (25% of all PTB), preterm premature rupture of the membranes (PPROM; 25% of all PTB), and spontaneous preterm labor (50% of all PTB) [6].

One pathophysiologic mechanism associated with spontaneous PTB is also associated with an imbalance between ROS and antioxidant defenses, in other words, oxidative stress. Oxidative stress induces DNA damage and telomere shortening, which accelerates telomere-dependent senescence of the fetal membranes and results in senescence-associated inflammatory activation that may contribute to parturition [67]. Spontaneous preterm labor or PPROM is likely to be triggered by premature placental aging caused by oxidative stress-induced damage and premature senescence of the intrauterine tissues, especially the fetal membranes of the placenta [2,68]. Intrauterine infection and inflammation, which are related to oxidative stress, are the main etiological factors in the pathogenesis of spontaneous PTB and risk factors for brain damage in prematurely born neonates [69].

Of course, an appropriate amount of oxidative stress in the placenta might be necessary for its development since it regulates trophoblast proliferation, differentiation, and invasion, promotes placental angiogenesis, and regulates the autophagy and apoptosis required for normal placentation. However, if oxidative stress levels are much higher than usual, PTB may occur more frequently. It is noteworthy that increased oxidative stress levels in pregnant women could cause placental dysfunction or other damage, inducing PTB, and be responsible for complications in prematurely born neonates because of the direct exchange of metabolites within the placenta. Some review articles reported an association between levels of ROS, RNS, or other by-products of oxidative stress (i.e., biomarkers) and prematurity and stated that the oxidative stress level was usually higher in PTB specimens than in term birth specimens. The antioxidant levels were also reportedly lower in PTB specimens than term birth specimens [34,70]. These results show an imbalance in oxidative stress and antioxidants in PTB; thus, the pathophysiology of PTB is associated with a redox dysregulation.

## 7. Antioxidants

Antioxidants prevent free radical-induced tissue damage by preventing the formation of scavenging radicals or by promoting their decomposition. To counterbalance ROS, cells have endogenous antioxidant systems that include non-enzymes, enzymes, and trace elements [71]. Antioxidants inhibit oxidant attacks on proteins, lipids, carbohydrates, and DNA in different ways. The most common enzymatic antioxidant, SOD, converts superoxide to H₂O₂ in both the mitochondria and cytosol. The impaired action of this enzyme or its associated enzymes has been linked to oxidative stress [72]. H₂O₂ can then be converted to H₂O by other enzymes such as catalase, peroxiredoxins, and/or glutathione peroxidase. These enzymes are extremely important for maintaining the delicate balance between superoxide and H₂O₂ because an increase in the latter may lead to reactions with endogenous transition metals via Fenton mechanics, resulting in the formation of hydroxyl ions. Although not mentioned in the ROS section above, these hydroxyl ions are among the most damaging free radicals in the human body. Furthermore, animal studies have revealed that the overexpression of enzymatic antioxidants, such as catalase, may have a protective effect against neurologic sequelae stemming from proton-induced irradiation [73,74].

Non-enzymatic antioxidants include ascorbate (vitamin C), tocopherols such as α-tocopherol (vitamin E), and carotenoids such as β-carotene, all of which scavenge and neutralize free radicals. Notably, antioxidant enzymes depend upon metallic co-factors, in other words, trace elements, which are capable of taking on different valencies since they transfer electrons during redox reactions; thus, metals such as copper, zinc, manganese, iron, and selenium may also be considered under the antioxidant umbrella. Polyphenols, which are found in a number of fruits and vegetables also serve as natural antioxidants [75].

α-lipoic acid (ALA) is a sulfur-containing fatty acid that plays a basic role as an antioxidant and is required for mitochondrial α-ketoacid dehydrogenase complex. ALA synthesis occurs enzymatically from octanoic acid in the mitochondria and is produced by plants and animals. However, its synthesis in humans is comparatively very low [76], so it must be taken by food, such as potatoes, broccoli, spinach, tomatoes, brown rice and red meat [77]. Its antioxidant propertied are attributed to several reasons including its capacity for direct scavenging for ROS, regeneration of endogenous antioxidants such as glutathione, vitamins C and E, and its metal-chelating activity [78]. This molecule is continuously introduced in our body with diet, also during pregnancy. No relevant side effects have been reported when ALA is taken with food or as dietary supplement, even when administered at much higher doses with respect to those used in normal treatments. Therefore, administration of ALA to pregnant women is regarded as safe under strict medical supervision [79].

### Potential Antioxidant-Based Therapy for Adverse Pregnancy Outcomes

Of course, the definite treatment for PE is delivery and termination of pregnancy, but prolonged pregnancy is sometimes necessary for fetal growth and maturation. Therefore, therapies that increase maternal antioxidant status have been receiving attention, and vitamin C and E supplementation is rising as an alternative treatment for PE. Such supplementation may even prevent PE [80]. Recent theories have reinforced the concept that antioxidants, such as ALA, can protect the fetus against oxidative stress, particularly toward the end of pregnancy [81]. ALA has a low molecular weight (206.32 Da) and can cross biological membranes, including the blood–brain barrier. Some studies demonstrated that ALA acted as an effective neuroprotector in mice models [82,83]. Sharma et al. [84] reported that lycopene (a bright red carotenoid) can decrease both PE and FGR risks, but Antartani et al. [85] found that only FGR risk was decreased. In addition, according to Rytlewsky et al. [86] L-arginine supplementation can accelerate fetal weight gain and improve biophysical profile in PE. However, some studies indicated that oxidative stress might be crucial only in some groups of pregnant women; therefore, there are no obvious benefits to using antioxidants to prevent such pathologies in all pregnant women. Another randomized clinical study failed to prove that vitamin C and E supplementation could positively affect the occurrence of PE [87]. The FACT trial (high-dose folic acid supplementation throughout pregnancy for preeclampsia prevention), a randomized double-blind placebo-controlled efficacy study recently reported no benefit of high-dose folic acid supplementation beyond the first trimester for the prevention of PE or related adverse maternal and neonatal outcomes [88]. In fact, some studies reported that it can even negatively impact cellular function in the trophoblast layer of the placenta. Overall, the interpretations offered to date about the effectiveness of antioxidants, such as vitamins, for decreasing the risk of PE have not been convincing. Nevertheless, future studies evaluating the effects of antioxidants other than vitamins C and E on decreasing the risk of complications during pregnancy, such as small for gestational age and low birth weight, are still warranted [89].

Only a few clinical studies to date have evaluated the potential benefits of antioxidants in GDM. Suhail et al. [90] reported that the levels of antioxidants such as selenium, zinc, and vitamin E were reduced in GDM. In addition, a diet rich in antioxidants promotes better health, while enhancing one’s total antioxidant status [61,91,92]. Short-term treatment with ALA in patients with type 2 diabetes has been suggested to improve lipid profile by improving oxidative stress and inflammatory responses. There is a clinical trial study which reported the curative effect of ALA in GDM pregnancies. In this study, women with GDM showed a significant decrease in the maternal circulating values of triglyceride-glucose index, triglyceride (TG), TG/high-density lipoprotein cholesterol, atherogenic index of plasma, and thiobarbituric acid-reactive substances in the ALA group compared to the placebo group at the end of the intervention [93].

Another study elucidated the association between lower antioxidant levels in early pregnancy and PTB. The results suggest that while the effects of antioxidants are not that significant, a definite association exists between them, which should be cautiously interpreted prior to the clinical application of antioxidants in early pregnancy for the prevention of PTB [94]. In addition, an analysis of amniotic fluid samples collected by transabdominal amniocentesis in women with PPROM demonstrated decreased total antioxidant capacity, but no difference in oxidative stress markers regardless of whether microbial invasion or frank chorioamnionitis was present [95]. ALA is also effective in treatment of PTB. G. Vitrano et al. [96] suggested that the combined administration with ALA by oral and vaginal route obtained a statistically significant improvement of symptoms, with a reduced cervical shortening in patients at risk for PTB. No adverse effects were detected during the treatment. The success of this new treatment would seem positively linked to a history of previous PTB, smoking and high maternal BMI.

Some other studies suggested that restoring mitochondrial fitness could be a potent therapy for adverse pregnancy outcomes [97]. A few cytoprotective agents target the mitochondria, such as MitoTEMPO (a mitochondria-targeted SOD antioxidant mimetic) and MitoQ (an orally active antioxidant consisting of a lipophilic triphenylphosphonium cation and coenzyme Q10), resulting in a significant reduction in cellular superoxide production, normalization of mitochondrial function, and reduction of inflammatory influx [98,99,100]. MitoTempo significantly reduced mitochondrial ROS generation in cells exposed to plasma from women with PE. Mitochondrial targeted antioxidant treatment was more effective than general antioxidant at similar concentrations, highlighting the importance of a direct-targeted therapeutic approach [101]. MitoQ also is 100-fold more potent than untargeted antioxidants in blocking ROS and preventing mitochondrial oxidative damage [102]. Yang et al. reported that MitoQ can protect against hypertension and kidney damage induced by reduced uterine perfusion pressure in mice with late gestation [103].

## 8. Conclusions

Pregnancy is a process of constant adaptation of the surrounding environment, and oxidative stress occurs mainly as a result of the high metabolic activity in the feto-placental compartment.

ROS levels are higher in PE, FGR, GDM and PTB than in normal pregnancies, and the oxidative stress level is related to disease course and severity. In addition, ROS overproduction can result in abnormal placentation and severe cellular damage, by destroying the normal protein, lipid, and DNA structures, leading to apoptotic changes within the placenta. It is possible that the management of oxidative stress could be beneficial, both prenatally and perinatally in women who are at risk of developing these diseases (Figure 1). 

ALA is rising as a good treatment option for adverse pregnancy outcomes, but it remains unclear whether antioxidant supplementation or an antioxidant-rich diet can improve the negative consequences of oxidative stress. Mitochondria targeted antioxidant treatment is also considered as a potent therapeutic approach. Clearly, further studies are required to better understand of the short- and long-term health benefits of reducing oxidative stress during pregnancy and to develop valid treatments for the adverse pregnancy outcomes.

## Figures and Tables

**Figure 1 ijms-22-10122-f001:**
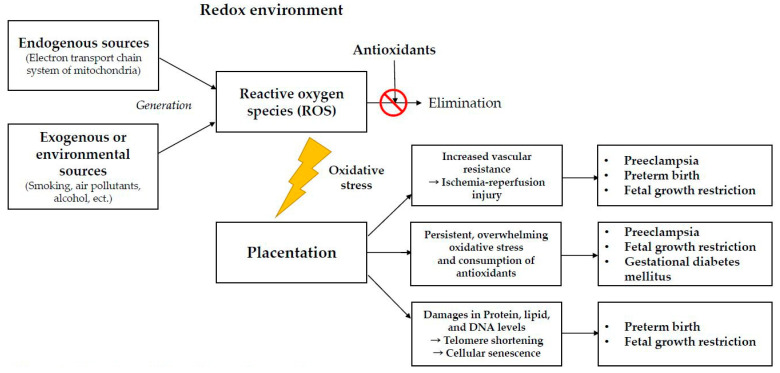
Overveiw of the redox environment. In the redox environment, ROS concentration can remain stable without damaging other organ systems. However, if the antioxidant-ROS balance is disrupted, oxidative stress will occur and induce abnormal placentation by increasing vascular resistance within the placenta, allowing oxidative stress to overwhelm antioxidants, and shortening telomeres, leading to cellular senescence. Preeclampsia, fetal growth restriction, gestational diabetes mellitus and preterm birth are the related complications.

## Data Availability

Data sharing not applicable.

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
