# Peer review of "Effect of Endogenic and Exogenic Oxidative Stress Triggers on Adverse Pregnancy Outcomes: Preeclampsia, Fetal Growth Restriction, Gestational Diabetes Mellitus and Preterm Birth"

_ijms, 2021, doi:10.3390/ijms221810122_

Round 1

Reviewer 1 Report

The authors responded to my comments.

Author Response

Thank you for your kind response. 

Reviewer 2 Report

Oxidative stress's role in pregnancy's bad course is already known for some time.  Impairment of the physiological mechanisms in early pregnancy has a deep impact, mostly expressed in the third trimester. Many pregnancy complications - preeclampsia, fetal growth restriction, gestational diabetes mellitus, and pre-term birth are attributable to oxidative stress. 

The article is well written and covers comprehensively known and poorly understood aspects of free radical's role in pregnancy's course. I have one suggestion for subsection 7.1. Sufficient data is available to support alpha-lipoic acid as a strong antioxidant, and, even more, there are studies to attest its role in high-risk pregnancies. The authors should include this therapeutic tool as an option for the management of pregnancy complications. 

Author Response

Thank you for your detailed comments. I added some information about alpha-lipoic acid treatment in adverse pregnancy outcomes as you commented. 

Please check this change and I'm looking forward to your new opinion. 

Reviewer 3 Report

The review focuses on a topic that is not fully addressed in the literature. Therefore, each paragraph is short and the number of citations is low. Furthermore, the authors did not clarify how they selected the papers reported in the review. The paragraph on antioxidant  is limited to product name in some cases.

Author Response

Thank you for your detailed comments. I tried to enrich the paragraph on antioxidant with more expert knowledges and added new up-to-date references within 5 years. Please check this change and give me another advice if you have. I’m looking forward to hearing from you.

This manuscript is a resubmission of an earlier submission. The following is a list of the peer review reports and author responses from that submission.

Round 1

Reviewer 1 Report

In this review, the authors tried to summarize the available literature on oxidative stress during pregnancy and its relationship to adverse pregnancy outcomes.  It is a very interesting subject to cover, and the authors outlined the main subjects to cover to achieve their purpose. However, the manuscript in its current form is severely flawed.

This reviewer will provide general comments to outline what should be targeted in order to make it more suitable for readers interested in this subject:

  1. The manuscript should undergo extensive English language revision from top to bottom. This is strongly recommended.
  2. The title does not make sense the way it is written, but more importantly, it does not reflect the content of the review. It should be rewritten.
  3. The aim of this review article is not well described in the introduction section. It should be rewritten accordingly
  4. The conclusion should be shortened and should be rewritten in a way that answers the aim of the review. This way, readers get a take-home message that is easy to remember and reference.
  5. Finally, this reviewer believes that based on the same structure used in the current manuscript, this review work would benefit from revisiting other references including, but not limited to: basic science papers that studied oxidative stress, ways to measure ROS generation and metabolic activity in different tissues, and oxidative stress in pregnant animal models. 

Author Response

Firstly, Thank you for detailed advice about the manuscript.

I have checked and revised the content as you mentioned that changing the introduction and conclusion sections and added more details about studies related oxidative stress. English revision was also done, so please check it and give me feedback. I'm looking forward to hearing from you soon.

Reviewer 2 Report

The paper entitled " The effect of endogenic and exogenic oxidative stress triggers on pregnancy adverse outcomes: preterm delivery, preeclampsia and gestational diabetes mellitus" aims to figure out the association between excessive ROS is detrimental and associated with many pregnancy complications, such as preeclampsia (PE), preterm birth(PTB), fetal growth restriction, and the death of mothers in late pregnancy, by damaging normal DNA structure.

My comments and criticism about this paper are given below:

  1. In recent years, some comprehensive review articles about and oxidative stress related defective placental dysfunction are contributing factors for negative pregnancy outcomes have been published. The role of ROS and premature placental ageing in the pathophysiology of pregnancy complications were systematically summarized. (Sultana Z, Am J Reprod Immunol. 2017 77(5). doi: 10.1111/aji.12653. Manna S, et al. Oxid Med Cell Longev. 2019 22;2019:3095383. Simon-Szabo Z et al, Exp Ther Med 2021 Jul;22(1):771, etc).

In this review article, the authors focus on the association between oxidative stress and adverse outcomes of pregnancy, especially PE, PTB and gestational diabetes, and explore how to overcome this oxidative stress in those unfavorable conditions. A systematic review is necessary of this field. However, clearly, this paper is a bit off-topic for a review manuscript. Most works mentioned in this paper are topsy-turvy arrangement and loose structure. This problem needs to be addressed to validify the rationality of this review.

  1. The paper is not well organized as a review paper. Sections 1 should be the introduction section. Sections 2 and 3 are confusing and there is no reason for introducing OS and measurement in a separated section. Section 4 and subsequent subsection are the major part to on topic, the content is loose and unfocused. Sections 5 are also confusing for introducing antioxidants-based therapy for women with ROS attack no matter the patients with PE, PTB or GDM; even though we all know that their pathogenic mechanisms are different.
  2. This paper lacks sufficient latest references. Less than 15 research articles cited in this paper are published within the last 5 years.
  3. It’s also very confused that why a chart (Figure 1) that appeared suddenly shows some simple and rough mechanisms without corresponding explanations.
  4. This paper is poor written and not easy to read. Please proof read the manuscript for grammatical and formatting errors. Different verb tenses are used through the manuscript use always past tense.

Author Response

Firstly, Thank you for detailed advice about the manuscript.

I have checked and revised the content as you mentioned that reorganizing the whole sections and added latest references. English revision was also done, so please check it and give me feedback. I'm looking forward to hearing from you soon.

Reviewer 3 Report

General comments:

A better knowledge of the molecules mechanisms through which oxidative stress influences pregnancy and related complications would help in the management of pregnancy-associated diseases.  To achieve this purpose the authors should extensively modify the manuscript taking into account also below suggestions. Overall, the title is ambitious and manuscript information are not very thorough, and sections are not connected each other.

I would suggest to reorganize the manuscript taking into account that a literature review should summarize the latest data on a given topic, uncover areas in which more research is needed, and also draw conclusions and suggestions for future studies.

Specific comments:

- GDM is not mentioned in the abstract, please add it

-“Introduction”: this section completely lacks literature information regarding antioxidants, please add

-2.1 and 2.1 sections are not necessary -they give information from the same study (ref.11)- and can be part of the “oxidative stress” section. Since the review concerns oxidative stress in pregnancy the authors  should give more detailed information on the role of oxidative stress in different body’ districts ( e.g., placenta, plasma, fetal cord…..etc); it is possible to combine “oxidative stress” and “oxidative stress in pregnancy” into one section

-“Measurement of oxidative stress” needs to be improved. More detailed information about how and where (in which tissues? Plasma, placenta…etc...) the measurement is made

-The authors dedicate section 5 and 5.1 to antoxidants in pregnancy, but they did not mention them in the abstract, they should do it

- Pag.4, lanes 146-166; lots of information without references; and ref.16 is not specific to what they say

Author Response

Firstly, Thank you for detailed advice about the manuscript.

I have checked and revised the content as you mentioned that reorganizing the whole manuscript and added more details about measurement of oxidative stress. English revision was also done, so please check it and give me feedback. I'm looking forward to hearing from you soon.

Reviewer 4 Report

In the manuscript “The Effect of Endogenic and Exogenic Oxidative Stress triggers on pregnancy adverse outcomes: Preterm delivery, Preeclampsia and Gestational diabetes mellitus” submitted to International Journal of Molecular Sciences, the authors described the association between oxidative stress and adverse outcomes of pregnancy, especially preeclampsia (PE), preterm birth (PTB) and gestational diabetes, and how to overcome this oxidative stress in those unfavorable conditions.

This review was interesting and important in this field but this reviewer has several concerns.

Through the manuscript, cited references are older. The authors should cite earlier studies.

In addition, references in this review are too few.

Lines 207-229, The relationship between oxidative stress and PE should be specifically discussed more.

Lines 247-253, The location of this paragraph is inappropriate. This paragraph should transfer into another section.

In section 5 “Antioxidant”, The authors need to describe the role and localization of the enzymatic antioxidants in the placenta, because the enzymatic antioxidants are highly expressed in the endometrium.

The authors should discuss the molecular mechanisms by which oxidative stress including ROS exacerbate the complications of pregnancy.

It would help the readers to have an illustration of the association between the pathology of pregnancy complications and oxidative stress.

Author Response

Firstly, Thank you for detailed advice about the manuscript.

I have checked and revised the content as you mentioned that reorganizing the whole manuscript and added latest references. In addition, I described the relationship between oxidative stress and preeclampsia much more specifically. English revision was also done, so please check it and give me feedback. I'm looking forward to hearing from you soon.

Round 2

Reviewer 1 Report

Some content-related concerns were properly addressed. However, since this manuscript is not easy to follow due to a very bad use of English and grammar, this reviewer believes that content cannot be properly assessed until the main issue is not tackled. All additions/modifications included have the same language flaws as the prior version.

English language and style require revision by a professional native English speaker and editor before thoroughly judging content.

Reviewer 3 Report

Confirm my previous comments

Reviewer 4 Report

The authors responded to my comments.